**Investigation**

# Ploidy inference from single-cell data: application to human and mouse cell atlases

Fumihiko Takeuchi [ID] ,[1,2,3,]* Norihiro Kato [ID] [3,4]

[1]Baker Department of Cardiometabolic Health, Melbourne Medical School, The University of Melbourne, Melbourne, VIC 3010, Australia
[2]Systems Genomics Laboratory, Baker Heart and Diabetes Institute, Melbourne, VIC 3004, Australia
[3]Department of Gene Diagnostics and Therapeutics, Research Institute, National Center for Global Health and Medicine, Tokyo 162-8655, Japan
[4]Department of Clinical Genome Informatics, Graduate School of Medicine, The University of Tokyo, Tokyo 113-0033, Japan

*Corresponding author: Systems Genomics Laboratory, Baker Heart and Diabetes Institute, 75 Commercial Road, Melbourne VIC 3004, Australia.
Email: fumihiko@takeuchi.name

Ploidy is relevant to numerous biological phenomena, including development, metabolism, and tissue regeneration. Single-cell RNA-seq and other omics studies are revolutionizing our understanding of biology, yet they have largely overlooked ploidy. This is likely due to the additional assay step required for ploidy measurement. Here, we developed a statistical method to infer ploidy from single-cell ATAC-seq data, addressing this gap. When applied to data from human and mouse cell atlases, our method enabled systematic detection of polyploidy across diverse cell types. This method allows for the integration of ploidy analysis into single-cell studies. Additionally, this method can be adapted to detect the proliferating stage in the cell cycle and copy number variations in cancer cells. The software is implemented as the scPloidy package of the R software and is freely available from CRAN.

**Keywords:** ploidy; single-cell; single-nucleus; ATAC-seq; cell cycle; copy number variation; cancer

## Introduction

While the majority of human and mouse somatic cells is diploid, certain cell types, such as hepatocytes, cardiac myocytes, and megakaryocytes, exhibit higher levels of ploidy (Biesterfeld *et al.* 1994; Ugo 2007; Orr-Weaver 2015; Fang *et al.* 2022). Ploidy is relevant to various biological phenomena, including development, tissue regeneration, metabolism, and cancer. The ploidy level, defined as the number of chromosome sets contained within a cell or nucleus, is termed "polyploid" when exceeding 2 sets. Ploidy can be measured by the DNA content in multiples of the *C* value, which represents the amount of DNA in a haploid genome. For example, the nuclear DNA content in a diploid or tetraploid cell (in the $G_1$ phase) corresponds to 2C and 4C, respectively.

Historically, cell ploidy has been measured through microscopy or flow cytometry, observing stained DNA or telomeres. Although integrating ploidy into single-cell analysis is possible by adding a flow cytometry step before single-cell assays (Richter *et al.* 2021), such procedures are not widely adopted. Here, we propose a statistical algorithm to infer ploidy from standard single-cell ATAC-seq (scATAC-seq) or single-nucleus ATAC-seq (snATAC-seq) data. We also propose a score to detect proliferating cells.

Additionally, copy number variation (CNV), the alteration in DNA copy number between chromosomal regions within a cell, is closely related to ploidy. There exist single-cell CNV detection algorithms that utilize data from single-cell RNA-seq (scRNA-seq) (Li *et al.* 2020) or snATAC-seq (Nikolic *et al.* 2021; Moore and Yardımcı 2023; Ramakrishnan *et al.* 2023). We show that our method, originally developed for ploidy inference, can also be applied to CNV detection.

## Methods
### Probability distribution for the method of moments

In ATAC-seq and whole-genome sequencing, the nucleotide sequence of chromosomal fragments is determined. At any site on an autosome, the maximum number of fragments that can encompass the site is 2 for a diploid cell and 4 for a tetraploid cell. In this context, PCR duplicates, which are amplified from the same chromosomal fragment, are considered identical. Our probabilistic model is based on this simple counting of fragments at a site. We assume that the fragments originate from a *p*-ploid cell in single-cell assays or from a *p*-ploid nucleus in single-nucleus assays.

At 1 site on an autosome, the number of sequenced fragments should follow a binomial distribution with *p* trials. The probability of observing *x* fragments is

$$\frac{p!}{x!(p-x)!} s^x (1-s)^{p-x}, \quad x = 0, \ 1, \ \ldots, \ p,$$

where *s* represents the success probability, in this case, the probability of sequencing this particular site within 1 chromosome. Observing across all sites on the genome, we tally the number of sites where *x* fragments were observed and denote this by $f_x$. We define

$$T_1 = \sum_{x > 0} x f_x, \tag{1}$$

$$T_2 = \sum_{x > 0} x^2 f_x. \tag{2}$$

The statistics $T_1$ and $T_2$ equal the first and second sample moments, respectively, multiplied by $n$, the total number of sites. Thus, the expectations are

$$E[T_1] = n\, p\, s,$$
$$E[T_2] = n\, p\, s\, \{(p-1)s + 1\}.$$

Since

$$E[T_2]/E[T_1] - 1 = (p-1)s, \tag{3}$$

we can use $U = T_2/T_1 - 1$ as an estimator of $(p-1)s$ as shown by Rider (Rider 1955). This estimator provides information about the ploidy of each cell, multiplied by an unknown constant $s$ that is uniform across cells.

The aforementioned model, however, is practically suboptimal for 2 reasons: (1) observations from adjacent sites are statistically dependent since typically 50–100 bp are sequenced together in a read, and (2) most sites would be covered by 0 fragments in the case of ATAC-seq. To address these issues, we alter the observation points from all sites on the genome to just the 5′ ends (on the positive strand) of sequenced fragments, thus obtaining $f_x$ as fragment-end depth count statistics (FEDECS). Consequently, we employ a truncated binomial distribution, excluding instances of 0 success. This modification does not affect the validity of the estimator $U$. First, $f_0$ is not used in the calculations of $T_1$ and $T_2$ (equations (1) and (2)). Second, even though the total number of observational points, $n$, remains unknown, it becomes irrelevant in equation (3) as it cancels out.

## Inferring ploidy using the method of moments

For the single cells obtained in an experiment, we compute $U$, which serves as an estimator of $(p-1)s$ (Fig. 1). In the case of a mixture of diploids, tetraploids, and octoploids, we expect $U$ to distribute in 3 clusters whose values maintain a ratio of $(2-1):(4-1):(8-1)$. We take the logarithm and, for robustness, restrict the range of values: the maximum value is capped at no more than twice the third quartile minus the median of the

original distribution, while the minimum value is no less than twice the first quartile minus the median. This adjusted value is denoted by $V_i$ for cell $i$.

Given the set $P$ of possible ploidies in a data set, we first estimate $\hat{s}$ that best fits the data:

$$\hat{s} = \underset{s}{\arg\min} \sum_i \min_{p \in P} \{V_i - \log(p-1) - \log(s)\}^2.$$

The estimated ploidy of cell $i$ is then calculated as

$$\hat{p}_i = \underset{p \in P}{\arg\min} \{V_i - \log(p-1) - \log(\hat{s})\}^2.$$

Alternatively, ploidy can be estimated allowing for a fractional value not necessarily in $P$ as

$$\hat{p}_i = U_i/\hat{s} + 1.$$

Although we do not further explore this, the fractional ploidy could potentially be utilized in inferring cell cycle phases.

## Inferring ploidy using expectation–maximization algorithm

Alternatively, we approach ploidy inference by considering FEDECS ($f_1$, $f_2$, $f_3$, …) as samples drawn from a mixture of categorical distributions. The source categorical distributions correspond to potential ploidy levels, and the algorithm assumes a fixed number of possible ploidies, for instance, 3 for diploids, tetraploids, and octoploids. We impose no restrictions on the categorical distributions, thereby being more permissive than the moment-based method. To assign each cell to a source distribution, we employ the expectation–maximization (EM) algorithm implemented in the mixtools package (version 1.2.0) (Benaglia et al. 2009) of the R software. Analysis of both simulated and empirical data indicates optimal performance when using only the counts ($f_2$, $f_3$, $f_4$), as $f_1$ is considerably larger and less informative than other components. If the EM algorithm fails to converge, possibly due to $f_2$ being disproportionately large compared with the

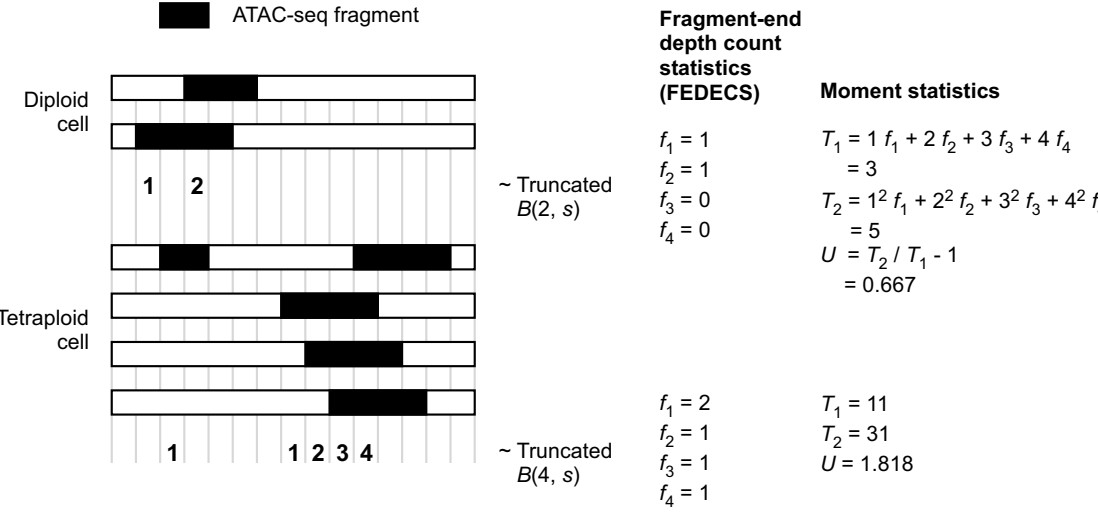

**Fig. 1.** Schematic representation of the probabilistic model employed in this study. At any site on an autosome, the number of snATAC-seq fragments encompassing the site from a diploid cell follows a binomial distribution with 2 trials and a success probability, $s$. For a tetraploid cell, the number of fragments follows a binomial distribution with 4 trials. Instead of observing all sites on the genome, we focus only on the 5′ ends (on the positive strand) of the fragments. Observations at relevant sites are highlighted in bold font.

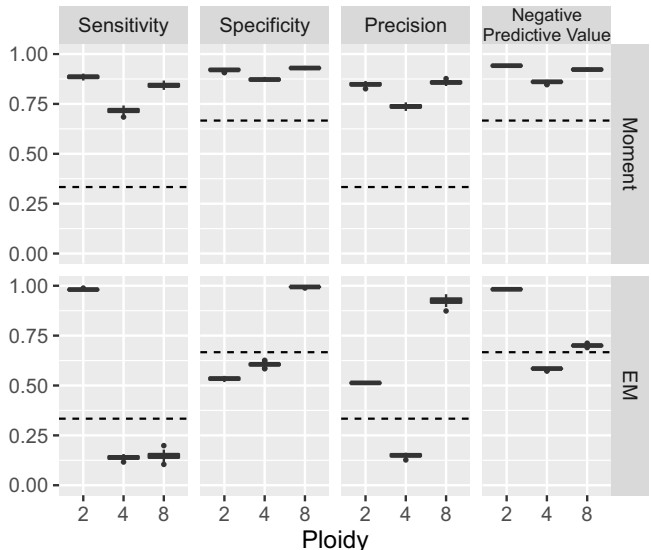

**Fig. 2.** Evaluation of ploidy inference algorithms using simulated data. Sensitivity, specificity, precision, and negative predictive value (vertical axis) for the prediction of diploids, tetraploids, and octoploids (columns) are plotted across 100 data sets simulated from an snATAC-seq experiment on human peripheral blood mononuclear cells. The upper and lower panels represent the method of moments and the EM algorithm, respectively. In the boxplots, the middle bar indicates the median, and the lower and upper hinges correspond to the first and third quartiles, respectively. The whiskers extend to no further than 1.5 times the interquartile range from the hinges, and data points beyond the hinges are plotted individually. Dashed lines indicate the performance of a random guess.

others, we opt to use ($f_3$, $f_4$, $f_5$) instead. Ploidy levels (e.g. 2, 4, and 8) are then assigned to the source distributions based on the size ordering of the parameter for the last category, which is $f_4$ when considering ($f_2$, $f_3$, $f_4$).

## Processing of DNA fragment data

Only fragments mapped to autosomes were retained, excluding those where the 5′ or 3′ end was located within simple repeats of the genome, as the mapping of next-generation sequencing reads in these regions can be imprecise. The simple repeats for both mouse and human genomes were sourced from the UCSC table browser (Karolchik *et al.* 2004). The FEDECS ($f_1$, $f_2$,..., $f_6$) was computed for each cell.

## Simulated data

To assess the ploidy inference algorithms, we generated simulated mixtures of diploid, tetraploid, and octoploid nuclei. snATAC-seq data from human peripheral blood mononuclear cells from a healthy male were downloaded from the 10x Genomics website (https://www.10xgenomics.com/resources/datasets/10-k-human-pbm-cs-multiome-v-1-0-chromium-controller-1-standard-2-0-0). The data set included 10,691 nuclei, each sequenced to an average of 57k read pairs. From each cell type, we randomly selected 7 nuclei repeatedly. One nucleus was retained as a diploid. ATAC-seq fragments from 2 other nuclei were combined to simulate a tetraploid nucleus. The remaining 4 nuclei were merged to simulate an octoploid nucleus. Each simulated experiment consisted of 1,521 diploids, tetraploids, and octoploids each, totaling 4,563 nuclei. We generated 100 such simulated experiments.

## Cell atlases for mouse and human

We applied ploidy inference to publicly available cell atlases of the mouse embryo (GSA CRA003910), human fetus (GEO GSE149683), adult mouse (GEO GSE111586), and adult human (GEO GSE184462). The data sets comprise 22, 60, 17, and 70 experiments with various tissue samples, respectively. We either downloaded the mapped snATAC-seq fragment data or computed it from FASTQ files using the Cell Ranger ATAC software (version 1.2.0). We retained nuclei with fragment counts ranging from $10^3$ to $10^5$. The total number of analyzed nuclei was 202,704 for the mouse embryo, 570,257 for the human fetus, 69,428 for the adult mouse, and 473,467 for the adult human. The cell type annotations from the data sets were standardized using the Cell Ontology (CL; version 2024 February 13) (Diehl *et al.* 2016).

For ploidy inference in cell atlases, we utilized the EM algorithm based on its superior negative predictive value for diploids in simulations, suggesting a higher likelihood of accurately predicting genuine polyploid cells. Additionally, the moment-based method, which presupposes a constant success probability $s$ of binomial distribution (defined in section Probability distribution for the method of moments), may not be universally applicable across all experiments within a data set. We assumed 3 possible ploidy classes (diploid, tetraploid, and octoploid) and computed ploidy inference for all nuclei in each of the 4 data sets.

Within each data set, we compared "organ–cell type" combinations and tested each for an excess of polyploid nuclei. We applied generalized linear regression with a binomial distribution, where the count of polyploid nuclei served as the dependent variable. By incorporating "organ–cell type" combinations as independent variables, we calculated the estimate and standard error for the proportion of polyploid nuclei in each combination. We also included experiment IDs as independent variables to adjust for batch effects. The background level, defined as the first quartile of estimates across all "organ–cell type" combinations, represents the proportion of polyploids observable even in diploid cell types. For each "organ–cell type," we assessed the excess over the background by calculating the $Z$-score and 1-sided $P$-value. Given 229 combinations tested across the data sets, we applied the Bonferroni correction, setting the significance level to $0.01/229 = 4.36 \times 10^{-5}$.

## Detecting proliferating cells

To identify actively proliferating cells, we estimate DNA copy number separately around replication origins and termini, assessing whether the former exceeds the latter. To be precise, given that our data are derived from ATAC-seq, we measure the number of open chromatin regions instead of DNA counts. However, the relative comparison between origins and termini remains valid. We define the surrounding region for each replication origin to extend from 1/4 the distance to the preceding origin to 1/4 the distance to the subsequent origin, with the distance capped at 50 kb if 1/4 exceeds this value. The outsides are defined as the surrounding regions for termini. In each cell, we computed the $U$ statistic (see above) for regions around origins and termini, denoted as $U_{ori}$ and $U_{ter}$, respectively. Given that the ratio of these 2 values remains constant in $G_1$ phase cells, regardless of ploidy, we employ linear regression with $U_{ori}$ as the dependent variable and $U_{ter}$ as an independent variable. The residual from this regression, representing the excess of $U_{ori}$, acts as an indicator of cell proliferation, which we designate the S phase score. To estimate the average of this score across cells within each "organ–cell type" combination, we introduce independent variables for "organ–cell type"

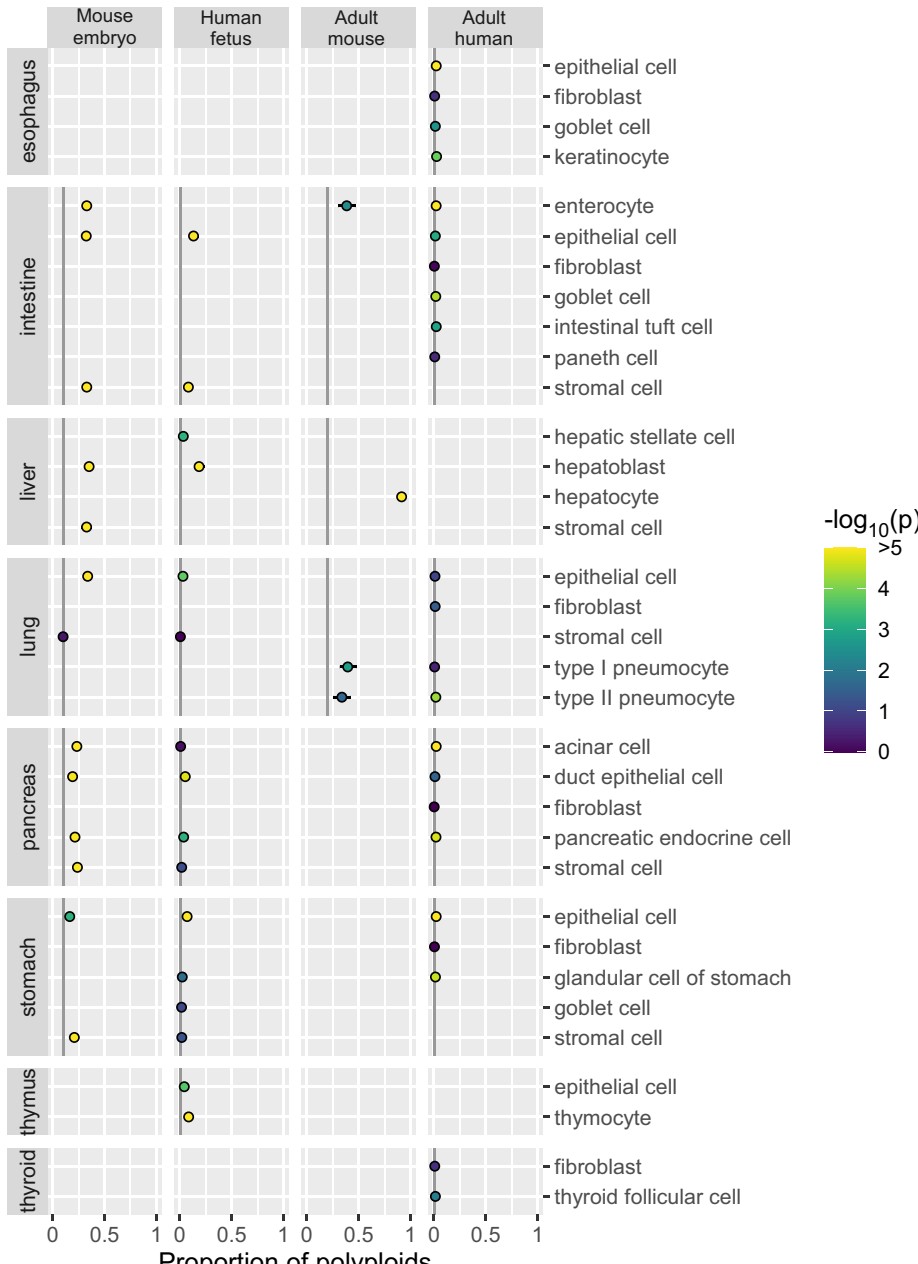

**Fig. 3.** Polyploidy of cell types in endoderm-derived organs. The plot is organized by cell atlas data sets (columns) and "organ–cell type" combinations (rows). The estimate and standard error of the proportion of polyploid nuclei are represented by a dot and a whisker. In each data set, the background proportion of polyploid nuclei detectable in diploid cell types is indicated by gray vertical lines. The *P*-value for excess of polyploids over the background is indicated by the color of dots.

combinations into the regression. The "organ–cell type" whose estimate ranks at the first quartile is chosen as the baseline. We evaluate the departure from the baseline for each "organ–cell type" by computing the 2-sided *P*-value. We also include experiment IDs as covariates to adjust for potential biases.

DNA replication origins shared across multiple assays were sourced from Tian *et al.* (2024). The analysis of proliferating cells was exclusively performed for humans, for which replication origins have been determined with high precision. As the coefficient of determination of $U_{\text{ter}}$ in the linear regression was 0.92 for GSE149683 and only 0.40 for GSE184462, the latter, having a smaller number of ATAC-seq fragments per nucleus, was not pursued.

## Detecting CNV in cancer cells

The BAM file for snATAC-seq of the gastric cancer cell line SNU-601 was downloaded from Sequence Read Archive (SRA) (SRX9447058), and the mapped snATAC-seq fragments file for the basal cell carcinoma sample SU008_Tumor_Pre was downloaded from Gene Expression Omnibus (GEO) (GSE129785). All chromosomes were analyzed. We retained nuclei with fragment counts ranging from $5 \times 10^3$ to $10^{5.5}$. The total number of analyzed nuclei was 3,783 for SNU-601 and 747 for SU008_Tumor_Pre. The nuclei were assumed to be either diploid or tetraploid.

We detect CNVs in single cells by computing the *U* statistic (see above) for every 20 Mb window of a cell. The algorithm operates in 2 steps: (1) assigning copy number gain/loss to each cell–window

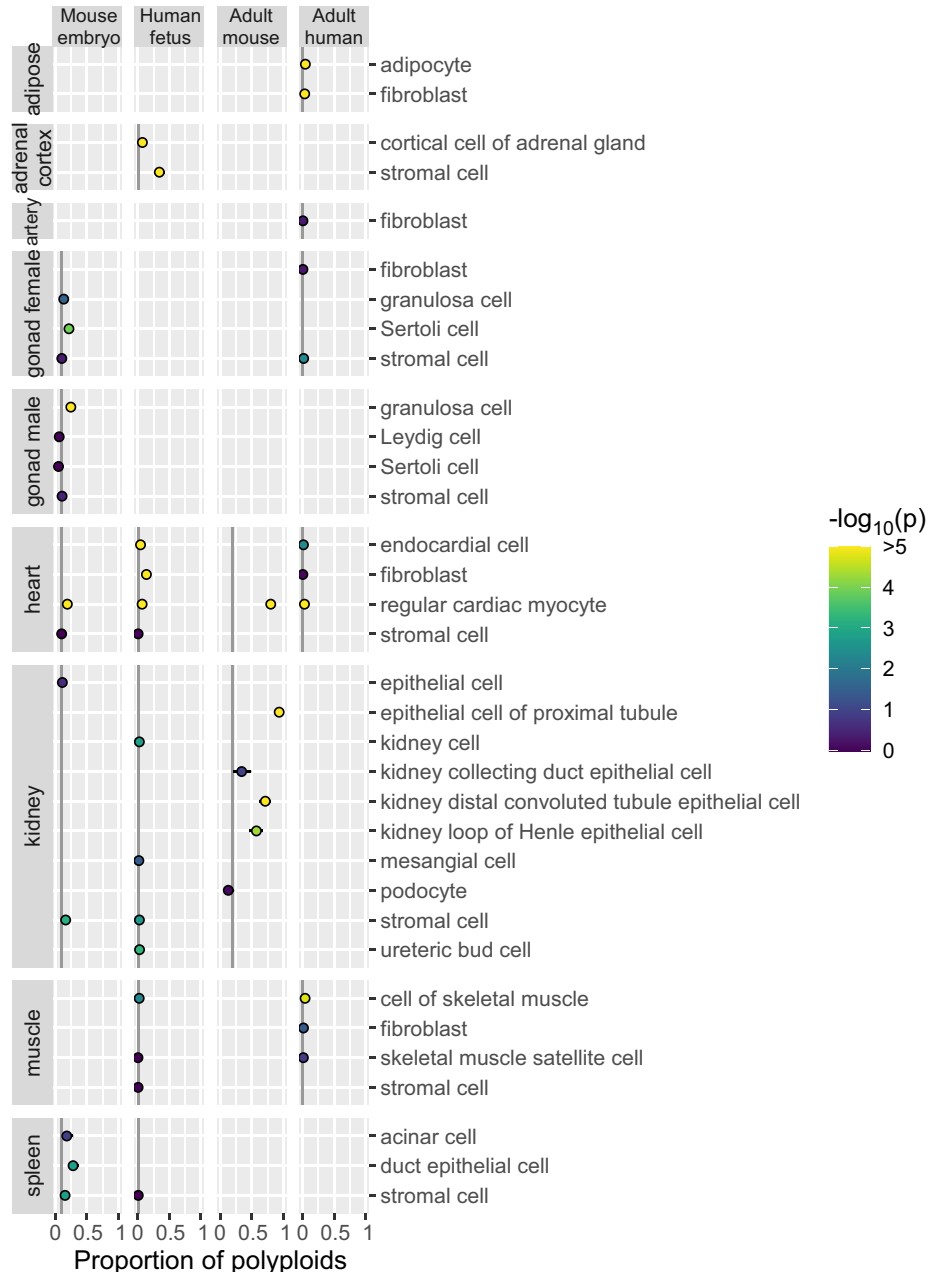

**Fig. 4.** Polyploidy of cell types in mesoderm-derived organs. Refer to the legend for Fig. 3.

and (2) integrating over all cells and windows to identify CNVs shared among a number of cells.

The raw $U$ statistic is computed for each cell–window and converted to $U'$ by correcting for chromatin accessibility, which influences the yield of ATAC-seq reads. For the subset of diploid cells, we conduct robust linear regression with $U$ as the dependent variable and multitissue chromatin accessibility (Meuleman *et al.* 2020) as the independent variable. The value of the independent variable varies by windows but remains constant across cells, as does the fitted value. The $U$ for each cell–window is divided by the fitted value and defined as $U'$. As $U$ is an estimator for $s$ in diploid cells, for which the robust linear regression is fitted, we can regard the conversion as division by $s$ yielding $U' = U/s$ on top of adjustment for chromatin accessibility.

Next, we compute the "ploidy" (more accurately, DNA copy number) of each cell–window after normalizing (or scaling) as if

the cell were diploid. We denote the true cell ploidy, true cell–window "ploidy," and the normalized cell–window "ploidy" as $p_{\text{cell}}$, $p_{\text{cell, window}}$, and $\tilde{p}_{\text{cell, window}}$, respectively. A gain or loss is defined when $\tilde{p}_{\text{cell, window}} \geq 3$ or $\leq 1$, respectively. With $U'_{\text{cell}}$ being the median of $U'_{\text{cell, window}}$ across windows in the cell, we compute the normalized cell–window ploidy as

$$\begin{aligned}
\tilde{p}_{\text{cell, window}} &= 2p_{\text{cell, window}}/p_{\text{cell}} \\
&= 2(U_{\text{cell, window}}/s + 1)/(U_{\text{cell}}/s + 1) \\
&= 2(U'_{\text{cell, window}} + 1)/(U'_{\text{cell}} + 1).
\end{aligned}$$

The second equation derives from $p = U/s + 1$.

Finally, we identify CNVs shared across a number of cells based on the gain/loss in cell–windows. For gains (and similarly for losses), occurrences are encoded in a 0/1 matrix of shape, cells ×

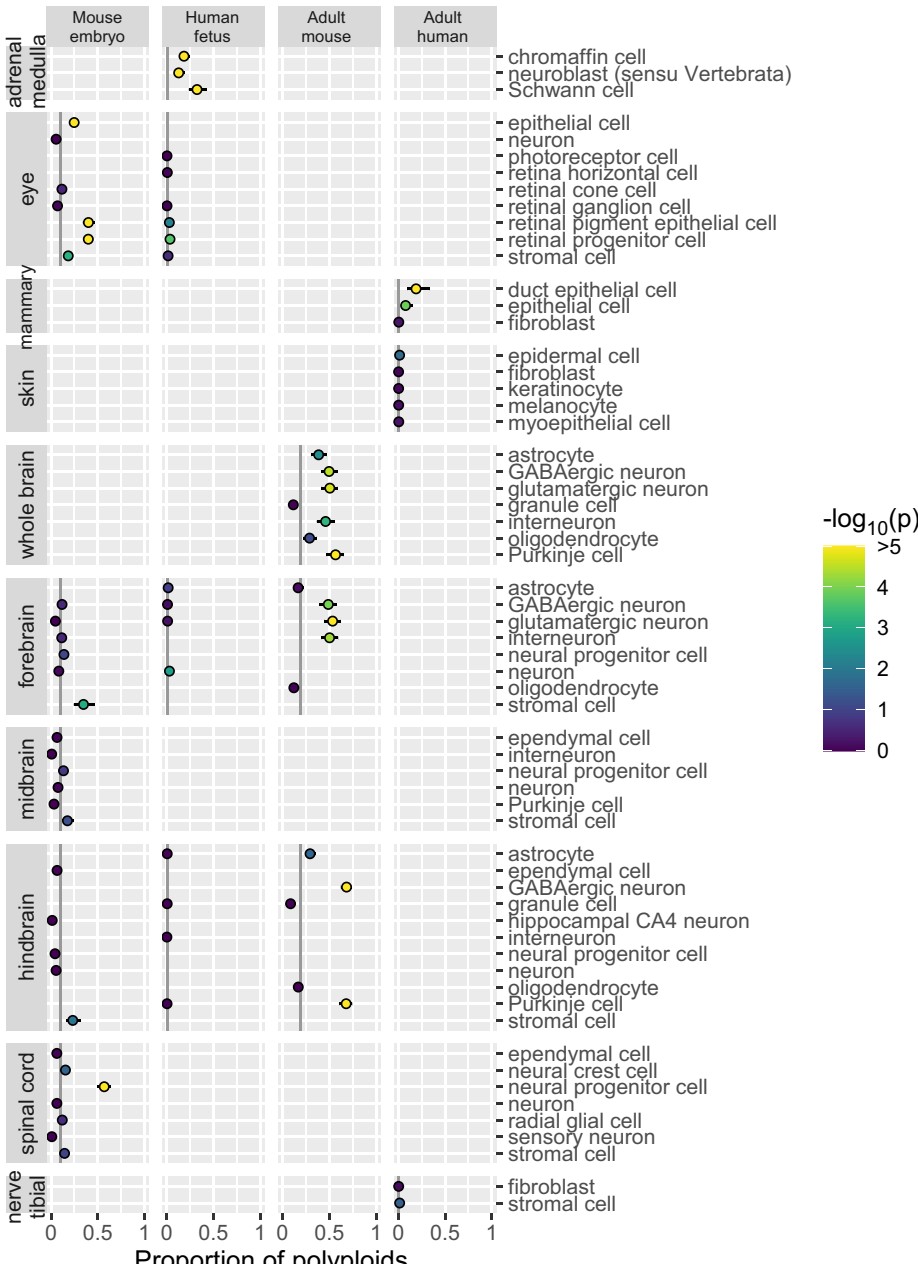

**Fig. 5.** Polyploidy of cell types in ectoderm-derived organs. Refer to the legend for Fig. 3.

windows. A series of (more than 1) consecutive windows exhibiting a high proportion of 1's in a subset of cells is identified as a CNV. The plausibility of a CNV is evaluated by the reduction in Bayesian information criteria (BIC) from the entire 0/1 matrix to the sub-matrices split between (subset of cells) × (series of windows) and the remaining. The ΔBIC is calculated for all possible series separately for gains and losses. CNVs are selected in a greedy manner starting from those with the smallest ΔBIC, ensuring that the series of subsequently chosen CNVs do not overlap with those previously selected.

## Results
### Accuracy of ploidy inference

In an scATAC-seq assay, when focusing on a single site on an autosome, up to 2 fragments encompassing the site can be observed from a diploid cell and up to 4 fragments from a tetraploid cell. This principle similarly applies to diploid and tetraploid nuclei within the context of an snATAC-seq assay. From a $p$-ploid cell, the number of fragments observable at a single site follows a binomial distribution with $p$ trials. To avoid redundant observations, we only consider the 5′ ends (on the positive strand of the genome) of ATAC-seq fragments instead of all sites on the genome, thereby sampling from a binomial distribution that is truncated to 1 or more successes (Fig. 1). We count the total number of 5′ ends across the genome, stratified by the number of encompassing fragments (i.e. depth), to derive the FEDECS for each cell.

We developed 2 algorithms to estimate ploidy from FEDECS. The first employs the method of moments for the aforementioned probability distribution and incorporates ploidy as a parameter. In contrast, the second algorithm models FEDECS as originating from a mixture of distinct categorical distributions each

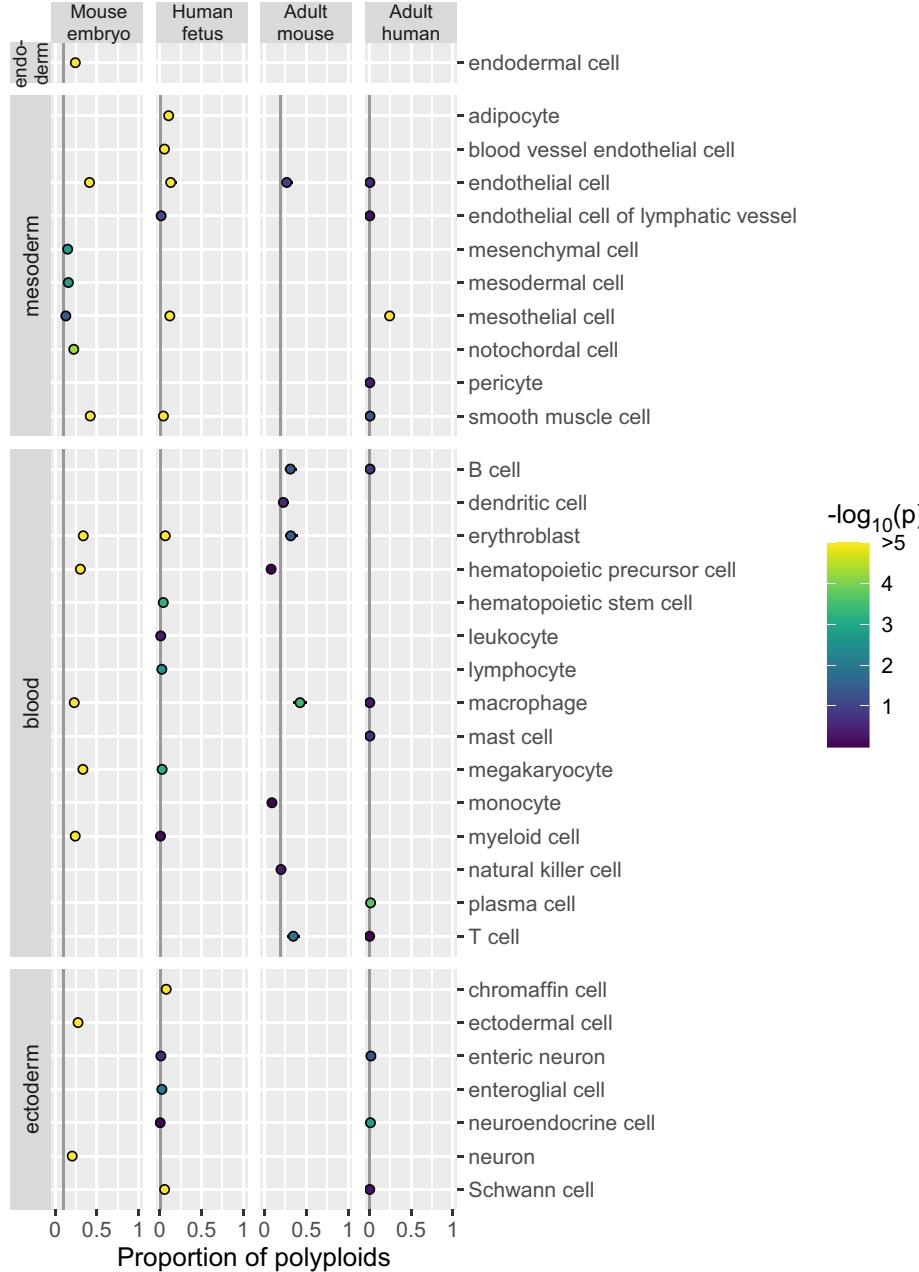

**Fig. 6.** Polyploidy of cell types that migrate or circulate to various organs. Refer to the legend for Fig. 3.

corresponding to possible ploidy levels (such as diploid, tetraploid, and octoploid) and infers the cellular affiliation using the EM algorithm. The first algorithm may perform better if the binomial distribution assumption holds true, whereas the second may excel if this assumption does not hold.

Our evaluation of these algorithms utilized simulated data, incorporating actual snATAC-seq data from human peripheral blood mononuclear cells (all diploid) and generating tetraploid or octoploid nuclei by merging 2 or 4 diploid nuclei, respectively. We created a mixture containing 1,521 instances each of diploid, tetraploid, and octoploid nuclei, totaling 4,563 nuclei. The algorithms' performance was assessed across 100 randomly generated data sets (Fig. 2). Sensitivity, specificity, precision, and negative predictive value for diploids, tetraploids, and octoploids were all high when applying the method of moments. In contrast,

the EM algorithm displayed some metrics inferior to those expected from random guesses. Notably, regarding the negative predictive value for diploids, the EM algorithm (mean = 0.982, SD = 0.0025) surpassed the moment method (mean = 0.941, SD = 0.0033), indicating higher accuracy when predicting polyploid (i.e. nondiploid) nuclei.

The performance of these algorithms could diminish with a reduced total number of ATAC-seq fragments. To assess the impact of next-generation sequencing depth, we conducted simulations with half or a quarter of the original data's reads (57k read pairs per nucleus), observing only a marginal reduction in accuracy (Supplementary Fig. 1). Considering that 25k read pairs per nucleus is the general recommendation, it is unlikely that significant changes in the performance of the ploidy inference algorithms will occur within the typical ranges of snATAC-seq experiments.

**Table 1.** Polyploid cell types detected in mouse and human cell atlases.

| Organ | Cell type | Proportion of polyploid cells[a] | | | | Previous reports of polyploidy |
|---|---|---|---|---|---|---|
| | | Mouse embryo | Human fetus | Adult mouse | Adult human | |
| **Endoderm derived** | | | | | | |
| Intestine | Enterocyte | **0.33** | — | 0.38 | **0.02** | Drosophila (Ohlstein and Spradling 2006) |
| Intestine | Epithelial cell | **0.32** | **0.13** | — | 0.01 | Drosophila (Øvrebø and Edgar 2018) |
| Intestine | Stromal cell | **0.33** | **0.08** | — | — | |
| Liver | Hepatoblast | **0.35** | **0.19** | — | — | |
| Liver | Hepatocyte | — | — | **0.91** | — | Mammals (Orr-Weaver 2015) |
| Liver | Stromal cell | **0.33** | — | — | — | |
| Lung | Epithelial cell | **0.34** | 0.03 | — | 0.01 | Human (Hamada et al. 1989) |
| **Mesoderm derived** | | | | | | |
| Adrenal cortex | Stromal cell | — | **0.34** | — | — | |
| Heart | Regular cardiac myocyte | **0.19** | **0.07** | **0.80** | **0.03** | Mouse (Orr-Weaver 2015), human (Ugo 2007) |
| Kidney | Epithelial cell of proximal tubule | — | — | **0.93** | — | Mouse and human tubular cell (Chiara et al. 2022) |
| Kidney | Kidney distal convoluted tubule epithelial cell | — | — | **0.71** | — | Mouse and human tubular cell (Chiara et al. 2022) |
| Blood | Erythroblast | **0.34** | **0.06** | 0.31 | — | |
| Blood | Hematopoietic Precursor cell | **0.30** | — | 0.08 | — | |
| Blood | Megakaryocyte | **0.33** | 0.03 | — | — | Mammals (Orr-Weaver 2015) |
| Various organs | Endothelial cell | **0.41** | **0.13** | 0.27 | 0.01 | Human cell line (Wagner et al. 2001) |
| Various organs | Smooth muscle cell | **0.42** | 0.04 | — | 0.01 | Rat and human (McCrann et al. 2008) |
| **Ectoderm derived** | | | | | | |
| Adrenal medulla | Schwann cell | — | **0.32** | — | — | |
| Eye | Retinal pigment epithelial cell | **0.40** | 0.03 | — | — | Rodents and human (Ke et al. 2022; Novikova and Grigoryan 2020) |
| Eye | Retinal progenitor cell | **0.40** | 0.03 | — | — | |
| Whole brain | Glutamatergic neuron | — | — | **0.51** | — | Human neuron (Biesterfeld et al. 1994) |
| Whole brain | Purkinje cell | — | — | **0.57** | — | Human (Ugo 2007) |
| Forebrain | Glutamatergic neuron | 0.05 | 0.01 | **0.54** | — | Human neuron (Biesterfeld et al. 1994) |
| Hindbrain | GABAergic neuron | — | — | **0.68** | — | Human neuron (Biesterfeld et al. 1994) |
| Hindbrain | Purkinje cell | — | 0.00 | **0.68** | — | Human (Ugo 2007) |
| Spinal cord | Neural progenitor cell | **0.57** | — | — | — | |
| Ectoderm | Ectodermal cell | **0.28** | — | — | — | |

[a]Bold font indicates significant excess of polyploid cells ($P < 4.36 \times 10^{-5}$).

## Inferring ploidy in cell atlases

We applied ploidy inference to publicly available data from cell atlases of the mouse embryo (Jiang et al. 2023), human fetus (Domcke et al. 2020), adult mouse (Cusanovich et al. 2018), and adult human (Zhang et al. 2021). First, we inferred ploidy of each nucleus by utilizing the EM algorithm, separately by the atlases. Next, for 229 combinations of "organ–cell type," we computed the proportion of polyploid nuclei (Figs. 3–5 for endoderm-, mesoderm-, and ectoderm-derived organs, respectively; Fig. 6 for cell types migrating or circulating to various organs; Supplementary Table 1). We selected the first quartile in each atlas as the background proportion of polyploid nuclei detectable even in diploid cell types: 0.101, 0.010, 0.196, and 0.005 (indicated by gray vertical lines in the figures). We then assessed the excess of polyploid nuclei above the background levels (coloring dots by $P$-value). By further requiring the proportion to be >0.5 for adult mouse and >0.25 for the other data sets, we detected polyploids in 26 "organ–cell type" combinations (Table 1).

The polyploidy of mammalian somatic cells has been extensively studied in hepatocytes and cardiac myocytes and has also been documented in megakaryocytes, neurons, and Purkinje cells (Biesterfeld et al. 1994; Ugo 2007; Orr-Weaver 2015; Nandakumar et al. 2021; Fang et al. 2022). Our findings confirm polyploidy in these cell types. Sixteen of the 26 polyploid cell types identified in this study (62%) have been previously reported, thereby validating the accuracy of our algorithm. Polyploidy in pulmonary epithelial cells has been identified in both infant and adult humans (Hamada et al. 1989). Similarly, tubular cells in the kidney exhibit polyploidy in both mice and humans (Chiara et al. 2022). Human vascular endothelial cells have been shown to become polyploid during in vitro culture (Wagner et al. 2001). Additionally, polyploidy is observed in vascular smooth muscle cells (McCrann et al. 2008) and retinal pigment epithelial cells (Novikova and Grigoryan 2020; Ke et al. 2022) across both rodents and humans. Polyploidy in intestinal enterocytes (Ohlstein and Spradling 2006) and epithelial cells (Øvrebø and Edgar 2018) has been reported in Drosophila.

Among the previously unreported polyploid cell types, the 2 from the adrenal cortex and medulla were detected and only detected in the human fetal atlas. In human fetuses, the adrenal gland grows rapidly, reaching a relative size 10 to 20 times larger than that of the adult adrenal gland (Mesiano and Jaffe 1997). Active proliferation seems plausible also for other unreported polyploid cells: stromal cell, hepatoblast, erythroblast, hematopoietic precursor cell, retinal progenitor cell, neural progenitor cell, and ectodermal cell in the ectoderm germ layer. For a diploid

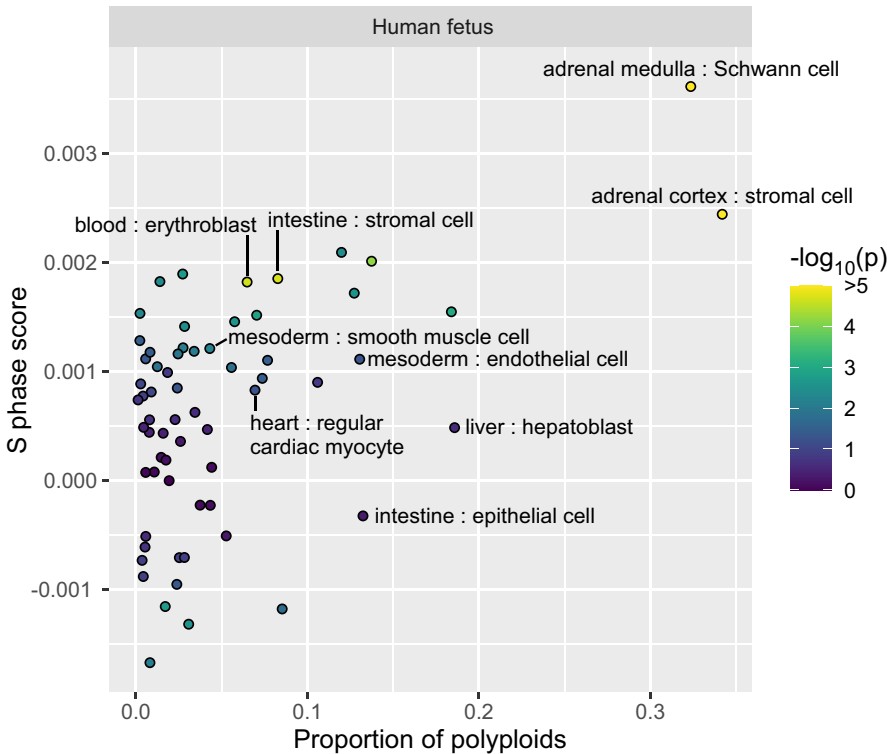

**Fig. 7.** Polyploidy and proliferation of cell types in the human fetal cell atlas. The proportion of polyploid cells (horizontal axis) and the S phase score (vertical axis) are plotted for each "organ–cell type" combination. Dot colors reflect the P-value for the S phase score being nonzero. The "organ–cell type" labels are indicated for combinations that were identified in this study and significant in the human fetal cell atlas (bold font in Table 1).

yet actively proliferating cell type, our algorithm predicts $G_1$ phase cells as diploid, $G_2$ phase cells as tetraploid, and S phase cells as intermediate in ploidy. We next investigate which cell types could be actively proliferating.

We refined the use of FEDECS to distinguish proliferating cells. The human genome contains ~50,000 origins of DNA replication (Tian *et al.* 2024). In the middle of the S phase, the DNA copy number is higher around replication origins (where replication has finished) and lower around replication termini (where replication has yet to start). In each cell, we computed FEDECS and inferred DNA copy numbers separately for origins and termini. The copy numbers are higher in tetraploid cells compared with diploid cells, but the ratio of origins to termini remains the same if the cells are in the $G_1$ phase. However, during the S phase, the value for origins becomes relatively higher. This excess at origins can serve as an indicator of cell proliferation, which we name the S phase score. By plotting the proportion of polyploids on the horizontal axis and the S phase score on the vertical axis, we expect proliferating cells to appear in the top right and nonproliferating polyploid cells in the bottom right (Fig. 7). In the human fetal cell atlas, active proliferation was detected in the abovementioned cell types in the adrenal gland, stromal cell in the intestine, and erythroblast in the blood. Furthermore, we identified polyploid cell types not engaged in active proliferation: cardiac myocyte in the heart, hepatoblast in the liver, and epithelial cell in the intestine.

### Detecting CNV in cancer cells

We detected CNVs in cancer cells by modifying the application of FEDECS. The genome was segmented into 20 Mb windows, and for each cell–window, we computed FEDECS and inferred copy number gains or losses. A CNV is defined as a consecutive series of windows showing a concentration of gains (or losses) within a

particular subset of cells. For the gastric cancer cell line SNU-601, our algorithm identified 12 gains and 8 losses in autosomes (Fig. 8a, Supplementary Table 2). We evaluated these results against a conservative gold standard—concordant detection by both Alleloscope (Wu *et al.* 2021) and epiAneufinder (Ramakrishnan *et al.* 2023) algorithms based on single-cell whole-genome sequencing. Our method yielded 13 true positives, 7 false positives, and 3 false negatives. The CNVs appeared uniformly across both diploid and tetraploid cells, with no evident clonal diversity. For the basal cell carcinoma sample SU008 (Satpathy *et al.* 2019), our approach detected 12 gains and 2 losses in autosomes (Fig. 8b, Supplementary Table 3). We compared these results against a conservative gold standard—CNVs detected concordantly in scRNA-seq and whole exome sequencing (Yost *et al.* 2019). Our algorithm yielded 8 true positives, 6 false positives, and 1 false negative. Notably, our algorithm has the capacity to uncover CNVs that do not occur uniformly across all cells, as demonstrated by gains at chr3q and chr6p. This tumor sample displayed clonal variation, with both clones comprising diploid and tetraploid cells.

### Discussion

Currently, the most widely used single-cell assays are RNA-seq, ATAC-seq, and their combined assay. Our ploidy inference algorithm is readily applicable to results from scATAC-seq and snATAC-seq without necessitating additional biochemical assays. Through simulations using real data, we demonstrated the high accuracy of our algorithm. Applying it to cell atlas data from humans and mice revealed a diverse range of polyploid cell types, corroborating previous reports on individual tissues. Although untested, theoretically, our algorithm could also be applied to single-cell whole-genome sequencing. This study represents a

**a** Gastric cancer cell line SNU-601

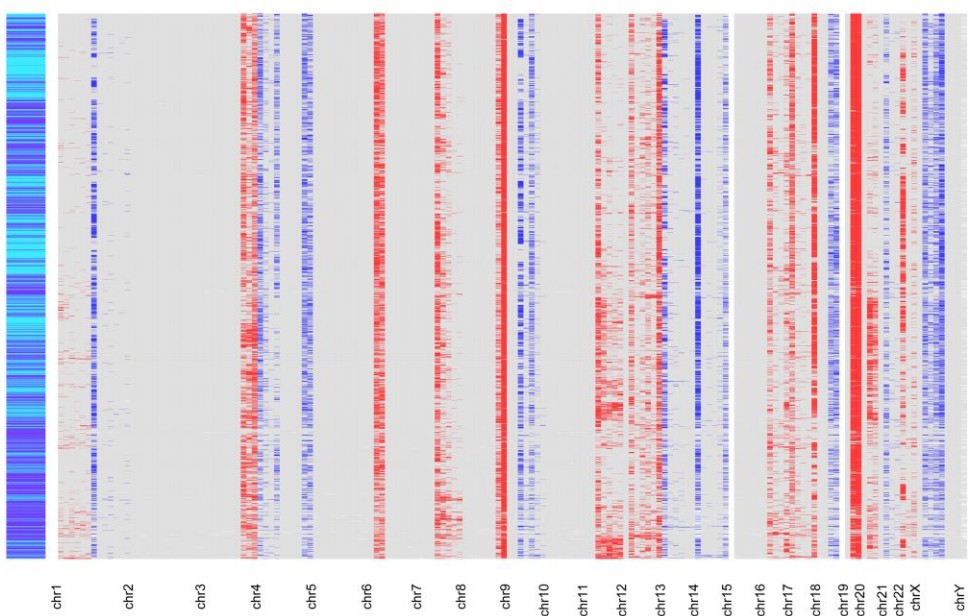

**b** Basal cell carcinoma sample SU008

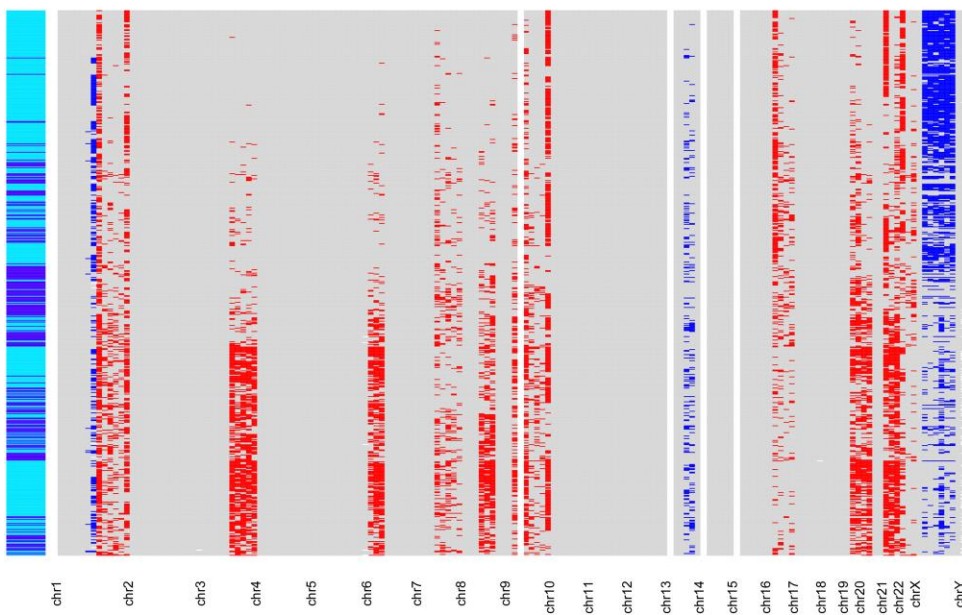

**Fig. 8.** CNV detected in cells from the gastric cancer cell line SNU-601 (a) and the basal cell carcinoma sample SU008 (b). In the heatmap, rows represent nuclei, and columns represent 20 Mb chromosomal windows. Each cell–window is color-coded: red for copy number gain within identified CNVs, blue for loss, gray for no change, and white for missing data due to insufficient ATAC-seq reads. The color bar on the left indicates the ploidy of nuclei, with diploids in dark blue and tetraploids in light blue.

pioneering systematic survey of polyploidy across a wide array of tissues. Integrating this algorithm into other single-cell studies could illuminate the relationship between ploidy and various biological phenomena at the single-cell level.

The ploidy inference was validated using in silico simulated data derived from actual snATAC-seq. However, in silico generated polyploids might differ from actual polyploids in terms of biochemical reactions, potentially leading to an overestimation of the algorithms' accuracy. Real validation could involve separating nuclei by ploidy using flow cytometry, indexing these nuclei, and

then pooling the suspension, though such data were not available to the authors. Among the 2 proposed algorithms, the method of moments assumes uniform sequencing efficiency across observed chromosomal sites, namely the 5′ ends of ATAC-seq fragments. Deviations from this assumption could reduce accuracy. Conversely, the EM algorithm, simply modeling a mixture of categorical distributions, could offer greater robustness.

We introduced the S phase score to measure single-cell proliferation by comparing DNA copy numbers at the origins and termini of replication. Several algorithms exist for assigning cell

cycle phases to single cells using scRNA-seq data (Andrews *et al.* 2021; Guo and Chen 2024). Our analysis is based on ATAC-seq and depicts the S phase without distinguishing between the $G_2$, M and $G_1$ phases. In the current work, we analyzed the proliferation levels averaged across cells within an "organ–cell type" combination. Future work will explore evaluating the S phase score at the single-cell level and integrating it with ploidy inference.

Cancer CNV in single cells has been inferred using RNA-seq or ATAC-seq data. Previous ATAC-seq-based methods simply quantified the total number of ATAC-seq fragments within a genomic window without considering fragment overlap (Nikolic *et al.* 2021; Moore and Yardımcı 2023; Ramakrishnan *et al.* 2023). We showed that FEDECS, which accounts for the overlap of fragments, can be utilized for CNV detection. FEDECS has the advantage of inferring both cell ploidy and CNV. However, its drawback is that it requires a larger number of fragments per window, leading to a coarser resolution of 20 Mb. In contrast, the simpler statistic allowed a previous algorithm to detect gains or losses in a cell–window with resolutions as fine as 250 kb (Moore and Yardımcı 2023), although this algorithm does not detect CNV as a series of windows.

## Data availability

The software is implemented as the scPloidy package of the R software and is freely available from CRAN. The source code is available on GitHub (https://github.com/fumi-github/scPloidy). The source code and data for the analysis in this article are available from figshare (https://doi.org/10.6084/m9.figshare.23574066). Publicly available snATAC-seq datasets were downloaded from Gene Expression Omnibus (https://www.ncbi.nlm.nih.gov/geo/), Genome Sequence Archive (https://ngdc.cncb.ac.cn/gsa/), and Sequence Read Archive (https://www.ncbi.nlm.nih.gov/sra/).

Supplemental material available at GENETICS online.

## Acknowledgments

This work utilized the supercomputing resource provided by the Human Genome Center (the University of Tokyo).

## Funding

This work was supported by the National Center for Global Health and Medicine Intramural Research Fund (20A1013).

## Conflicts of interest

The authors declare no conflicts of interest.

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

*Editor: H. Zhao*