## [Peer Review File · Genetics]

Ploidy inference from single-cell data: application to human and mouse cell atlases

Fumihiko Takeuchi and Norihiro Kato

NOTE: The reviews and decision letters are unedited and appear as submitted by the reviewers.

In extremely rare instances and as determined by a Senior Editor or the EIC, portions of a review may be redacted. If a review is signed, the reviewer has agreed to no longer remain anonymous.

The review history appears in chronological order.

Review Timeline:

Submission Date:	2023-08-23
Editorial Decision:	2023-09-27
Resubmission Received:	2024-03-18
Accepted:	2024-04-09

September 27, 2023

GENETICS-2023-306452

Ploidy inference from single-cell data: application to human and mouse cell atlases

Dear Dr. Takeuchi:

Two experts in the field have reviewed your manuscript, and I have read it as well. While your manuscript is not currently acceptable for publication in GENETICS, we would welcome a substantially revised manuscript. Both reviewers have comments and concerns to be addressed in a revised manuscript. You can read their reviews at the end of this email. Both reviewers have suggested you look at more data sets, discuss biological discoveries enabled by your method, and more downstream analysis. We look forward to receiving your revised manuscript. Please let the editorial office know approximately how long you expect to need for revisions.

Upon resubmission, please include:

1. A clean version of your manuscript;
2. A marked version of your manuscript in which you highlight significant revisions carried out in response to the major points raised by the editor/reviewers (track changes is acceptable if preferred);
3. A detailed response to the editor's/reviewers' feedback and to the concerns listed above. Please reference line numbers in this response to aid the editor and reviewers.

Your paper will likely be sent back out for review.

Additionally, please ensure that your resubmission is formatted for GENETICS
<https://academic.oup.com/genetics/pages/general-instructions>

Follow this link to submit the revised manuscript: Link Not Available

Sincerely,

Hongyu Zhao
Associate Editor
GENETICS

Approved by:
Sharon Browning
Senior Editor
GENETICS

Reviewer #1 (Comments for the Authors (Required)):

This manuscript introduces a smart method for inferring ploidy from single-cell ATAC-seq and single-nucleus ATAC-seq data. It has the potential to update our knowledge about polyploids because there are tons of newly generated single-cell data from the field.

However, there are a few areas where the manuscript could benefit from further improvement.

- 1) How reliable is the result? Is the method ready for making solid and new biological discoveries? If only the simulation data and some previously reported cell types mentioned in line 113 - line 125 support the polyploid results identified by the authors, it is possible that the method has an overfitting problem. How many cell types in total are newly identified as polyploids in this study? Could the author provide experimental or other independent evidence to support that at least one of those newly identified polyploid cell types is true? It is better to offer a new column in Table 1 to list the evidence, such as 'from literature', 'experimentally validated', and 'need further investigation'.
- 2) In Table 1, proliferating cells which are polyploids should be listed and discussed separately. Conceptually, it is less surprising. Another related question is, does the ploidy is only inferred at the cell type level? Could the author infer the polyploid in a single-cell level, or infer the proportion of polyploid cells in a given cell population or cell type?
- 3) Maybe it is too much to ask, but I think it will be very interesting to apply the author's method to snATAC data from tumor tissue to see what happens. Furthermore, is it possible to test whether a chromosome arm or a 100 MB window is polyploid?
- 4) It is worth noting that several 'Epithelial' cell types are listed in Table 1. However, seems the result tends to be stage-specific

or species-specific. On the other hand, almost every organ has epithelial cells. Could the author try to explain more about why some epithelial cells are special and some are just diploid as normal?

5) I suggest the author give a name to their method.

Addressing these points in the revision would significantly enhance the manuscript's overall quality and impact.

Reviewer #2 (Comments for the Authors (Required)):

Takeuchi et al. introduce a method to infer ploidy from single-cell ATAC-seq. This is an interesting angle to obtain information from single-cell data and it is potentially useful to improve our understanding of polyploidy in various biological systems. More real data examples and a comprehensive discussion of potential downstream analyses can further strengthen the manuscript. My detailed comments are as follows.

1. The authors have applied their method on the cell atlases data. As these data all contain multiple samples, it will be good if the authors can estimate and report the variability of the proportion of cells being polyploidy across different samples.
2. The atlases data are all from healthy samples. Tumor cells however are another type of cell that is known to have a large amount of variation in the amount of DNA. It will be good if the authors can explore their method in tumor samples and discuss the potential applications.
3. The cell cycle states are closely related to the ploidy calling. It will be good if the author can further discuss how the current method relates to different cell cycle stages.
4. The model is based on a few assumptions. It is unclear whether these assumptions are held in the real datasets.
5. It is unclear what test is conducted for Table 1 and whether it is meaningful to report such small p-value. The author can report the proportion of cells being polyploidy instead.
6. Is it possible to characterize the cells that are polyploidy in each cell type and identify the difference of these cells compared to the others?
7. Page 9 line 258-263, I am not sure which figure this sentence refers to.

Associate Editor Comments:

We thank the reviewers for his/her time and constructive comments, which we have incorporated into the revised manuscript.

Response to reviewer #1

This manuscript introduces a smart method for inferring ploidy from single-cell ATAC-seq and single-nucleus ATAC-seq data. It has the potential to update our knowledge about polyploids because there are tons of newly generated single-cell data from the field.

However, there are a few areas where the manuscript could benefit from further improvement.

1) How reliable is the result? Is the method ready for making solid and new biological discoveries? If only the simulation data and some previously reported cell types mentioned in line 113 - line 125 support the polyploid results identified by the authors, it is possible that the method has an overfitting problem.

We clarified the validation process and stated the limitations.

Discussion (Line 217): The ploidy inference was validated using in silico simulated data derived from actual snATAC-seq. However, in silico generated polyploids might differ from actual polyploids in terms of biochemical reactions, potentially leading to an overestimation of the algorithms' accuracy. Real validation could involve separating nuclei by ploidy using flow cytometry, indexing these nuclei, and then pooling the suspension, though such data were not available to the authors. Among the two proposed algorithms, the method of moments assumes uniform sequencing efficiency across observed chromosomal sites, namely the 5' ends of ATAC-seq fragments. Deviations from this assumption could reduce accuracy. Conversely, the EM algorithm, simply modeling a mixture of categorical distributions, could offer greater robustness.

How many cell types in total are newly identified as polyploids in this study? Could the author provide experimental or other independent evidence to support that at least one of those newly identified polyploid cell types is true? It is better to offer a new column in Table 1 to list the evidence, such as 'from literature', 'experimentally validated', and 'need further investigation'.

We detected polyploids in 26 cell types. Sixteen of the 26 (62%) have been previously reported. This study utilizes bioinformatics, and it was beyond the authors' expertise to perform wet lab experiment for validation. We added a column in Table 1 that describes the species in which polyploidy was reported and the reference.

2) In Table 1, proliferating cells which are polyploids should be listed and discussed separately. Conceptually, it is less surprising.

We realized that it is not obvious if cells of certain type are proliferating. To address this, we added new analysis in this revision.

Results (Line 163): We refined the use of FEDECS to distinguish proliferating cells. The human genome contains approximately 50,000 origins of DNA replication (Tian *et al.* 2024). In the middle of the S phase, the DNA copy number is higher around replication origins (where replication has finished) and lower around replication termini (where replication has yet to start). In each cell, we computed FEDECS and inferred DNA copy numbers separately for origins and termini. The copy numbers are higher in tetraploid cells compared to diploid cells, but the ratio of origins to termini remains the same if the cells are in the G₁ phase. However, during the S phase, the value for origins becomes relatively higher. This excess at origins can serve as an indicator of cell proliferation, which we name the S phase score. By plotting the proportion of polyploids on the horizontal axis and the S phase score on the vertical axis, we expect proliferating cells to appear in the top right and non-proliferating polyploid cells in the bottom right (Figure 7). In the human fetal cell atlas, active proliferation was detected in the above-mentioned cell types in adrenal gland, stromal cell in intestine, and erythroblast in blood. Furthermore, we identified polyploid cell types not engaged in active proliferation: cardiac myocyte in heart, hepatoblast in liver, and epithelial cell in intestine.

Figure 7. Polyploidy and proliferation of cell types in the human fetal cell atlas.

The proportion of polyploid cells (horizontal axis) and the S phase score (vertical axis) are plotted for each 'organ : cell type' combination. Dot colors reflect the P-value for the S phase score being non-zero.

Another related question is, does the ploidy is only inferred at the cell type level? Could the author infer the polyploid in a single-cell level, or infer the proportion of polyploid cells in a given cell population or cell type?

We added description for clarity.

Results (Line 123): We applied ploidy inference to publicly available data from cell atlases of the mouse embryo (Jiang *et al.* 2023), human fetus (Domcke *et al.* 2020), adult mouse (Cusanovich *et al.* 2018), and adult human (Zhang *et al.* 2021). First, we inferred ploidy of each nucleus by utilizing the EM algorithm, separately by the atlases. Next, for 229 combinations of 'organ : cell type' we computed the proportion of polyploid nuclei.

3) Maybe it is too much to ask, but I think it will be very interesting to apply the author's method to snATAC data from tumor tissue to see what happens. Furthermore, is it possible to test whether a chromosome arm or a 100 MB window is polyploid?

We added a new section for CNV detection in tumor cells.

Results (Line 182): Detecting CNV in cancer cells

We detected CNVs in cancer cells by modifying the application of FEDECS. The genome was segmented into 20 Mb windows, and for each cell-window, we computed FEDECS and inferred copy number gains or losses. A CNV is defined as a consecutive series of windows showing a concentration of gains (or losses) within a particular subset of cells. For the gastric cancer cell line SNU-601, our algorithm identified 12 gains and 8 losses in autosomes (Figure 8(a), Supplementary Table S2). We evaluated these results against a conservative gold standard—concordant detection by both Alleloscope (Wu *et al.* 2021) and epiAneufinder (Ramakrishnan *et al.* 2023) algorithms based on single-cell whole genome sequencing. Our method yielded 13 true positives, 7 false positives, and 3 false negatives. The CNVs appeared uniformly across both diploid and tetraploid cells, with no evident clonal diversity. For the basal cell carcinoma sample SU008, our approach detected 12 gains and 2 losses in autosomes (Figure 8(b), Supplementary Table S3). We compared these results against a conservative gold standard—CNVs detected concordantly in scRNA-seq and whole exome sequencing (Yost *et al.* 2019). Our algorithm yielded 8 true positives, 6 false positives, and 1 false negative. Notably, our algorithm has the capacity to uncover CNVs that do not occur uniformly across all cells, as demonstrated by gains at chr3q and chr6p. This tumor sample displayed clonal variation, with both clones comprising diploid and tetraploid cells.

(a) Gastric cancer cell line SNU-601

(b) Basal cell carcinoma sample SU008

Figure 8. CNV detected in cells from the gastric cancer cell line SNU-601 (a) and the basal cell carcinoma sample SU008 (b).

In the heatmap, rows represent nuclei, and columns represent 20 Mb chromosomal windows. Each cell-window is color-coded: red for copy number gain within identified CNVs, blue for loss, gray for no change, and white for missing data due to insufficient ATAC-seq reads. The color bar on the left indicates the ploidy of nuclei, with diploids in dark blue and tetraploids in light blue.

4) It is worth noting that several 'Epithelial' cell types are listed in Table 1. However, seems the result tends to be stage-specific or species-specific. On the other hand, almost every organ has epithelial cells. Could the author try to explain more about why some epithelial cells are special and some are just diploid as normal?

Epithelial cells overall tended to have high ploidy levels (Figures 3, 4, 5) with the five most prominent ones listed in Table 1. Those in Table 1 are found across developmental stages and in both mice and humans. We could not find characterization for the variation in polyploidy.

5) I suggest the author give a name to their method.

The newly developed statistic is now named fragment-end depth count statistics (FEDECS).

Addressing these points in the revision would significantly enhance the manuscript's overall quality and impact.

Response to Reviewer #2

Takeuchi et al. introduce a method to infer ploidy from single-cell ATAC-seq. This is an interesting angle to obtain information from single-cell data and it is potentially useful to improve our understanding of polyploid in various biological systems. More real data examples and a comprehensive discussion of potential downstream analyses can further strengthen the manuscript. My detailed comments are as follows.

1. The authors have applied their method on the cell atlases data. As these data all contain multiple samples, it will be good if the authors can estimate and report the variability of the proportion of cells being polyploidy across different samples.

We improved the methodology and added standard error for the proportion of polyploid cells as whiskers in Figures 3, 4, 5 and 6.

Methods (Line 368): Within each dataset, we compared 'organ : cell type' combinations and tested each for an excess of polyploid nuclei. We applied generalized linear regression with a binomial distribution, where the count of polyploid nuclei served as the dependent variable. By incorporating 'organ : cell type' combinations as independent variables, we calculated the estimate and standard error for the proportion of polyploid nuclei in each combination. We also included experiment IDs as independent variables to adjust for batch effects.

2. The atlases data are all from healthy samples. Tumor cells however are another type of cell that is known to have a large amount of variation in the amount of DNA. It will be good if the authors can explore their method in tumor samples and discuss the potential applications.

We added a new section for CNV detection in tumor cells. See reply to point 3) of reviewer #1 in Pages 3–4.

3. The cell cycle states are closely related to the ploidy calling. It will be good if the author can further discuss how the current method relates to different cell cycle stages.

We added a new analysis to detect S phase of cell cycle. See reply to point 2) of reviewer #1 in Page 2.

We also added explanation in the discussions:

Discussion (Line 228): We introduced the S phase score to measure single-cell proliferation by comparing DNA copy numbers at the origins and termini of replication. Several algorithms exist for assigning cell cycle phases to single cells using scRNA-seq data (Andrews *et al.* 2021; Guo and Chen 2024). Our analysis is based on ATAC-seq and depicts the S phase without distinguishing between the G₂, M and G₁ phases. In the current work, we analyzed the proliferation levels averaged across cells within an 'organ : cell

type' combination. Future work will explore evaluating the S phase score at the single-cell level and integrating it with ploidy inference.

4. The model is based on a few assumptions. It is unclear whether these assumptions are held in the real datasets.

We clarified the assumption and stated the limitations.

Discussion (Line 223): Among the two proposed algorithms, the method of moments assumes uniform sequencing efficiency across observed chromosomal sites, namely the 5' ends of ATAC-seq fragments. Deviations from this assumption could reduce accuracy. Conversely, the EM algorithm, simply modeling a mixture of categorical distributions, could offer greater robustness.

5. It is unclear what test is conducted for Table 1 and whether it is meaningful to report such small p-value. The author can report the proportion of cells being polyploidy instead.

We improved the analysis methodology and now report proportion of polyploids in Table 1, Figures 3, 4, 5 and 6, and Supplementary Table S1.

6. Is it possible to characterize the cells that are polyploidy in each cell type and identify the difference of these cells compared to the others?

Relation between ploidy and biological function is of interest to the authors but will be left as a topic of future work.

7. Page 9 line 258-263, I am not sure which figure this sentence refers to.

We added description.

Methods (Line 362): Additionally, the moment-based method, which presupposes a constant success probability s of binomial distribution (defined in section Probability distribution for the method of moments), may not be universally applicable across all experiments within a dataset.

April 9, 2024

RE: GENETICS-2024-306952

Dr. Fumihiko Takeuchi
The University of Melbourne
Baker Department of Cardiometabolic Health
Grattan Street
Parkville, N/A 3010
Australia

Dear Dr. Takeuchi:

Congratulations! We are delighted to inform you that your manuscript entitled "Ploidy inference from single-cell data: application to human and mouse cell atlases" is acceptable for publication in GENETICS. Both reviewers have found that your revised manuscript much improved and has addressed the previous comments. Many thanks for submitting your research to the journal.

To Proceed to Production:

1. Format your article according to GENETICS style, as discussed at <https://academic.oup.com/genetics/pages/general-instructions>, and upload your final files at <https://genetics.msubmit.net>.
2. Your manuscript will be published as-is (unedited-as submitted, reviewed, and accepted) at the GENETICS website as an Advanced Access article and deposited into PubMed shortly after receipt of source files and the completed license to publish. Please notify sourcefiles@thegsajournals.org if you do not wish to publish your article via Advanced Access.
3. We invite you to submit an original color figure related to your paper for consideration as cover art. Please email your submission to the editorial office or upload it with your final files. You can submit a small-sized image for evaluation, and if selected, the final image must be a TIFF file 2513px wide by 3263px high (8.375 by 10.875 inches; resolution of 600ppi). Please avoid graphs and small type.

If you have any questions or encounter any problems while uploading your accepted manuscript files, please email the editorial office at sourcefiles@thegsajournals.org.

Sincerely,

Hongyu Zhao
Senior Editor
GENETICS

Approved by:
Howard Lipshitz
Editor in Chief
GENETICS

note: Please add jnls.author.support@oup.com and genetics.oup@kwgglobal.com (or the domains @oup.com and @kwgglobal.com) to your email program's "safe senders" list. You will be contacted by both at various points during the production process.

Review comments (if applicable):

Reviewer #1 (Comments for the Authors (Required)):

The revised version has addressed my questions.

Reviewer #2 (Comments for the Authors (Required)):

I appreciate the authors' efforts in addressing all the concerns and questions raised in the first round of review. The authors have significantly improved the manuscript with the inclusion of several additional real data applications.